# BENCHMARKING APPROXIMATE k-NEAREST NEIGHBOUR SEARCH FOR BIG HIGH DIMENSIONAL DYNAMIC DATA

## ABSTRACT

Approximate k-Nearest Neighbour (ANN) methods are commonly used for mining information from big high-dimensional datasets. For each application the high-level dataset properties and run-time requirements determine which method will provide the most suitable tradeoffs. However, due to a significant lack of comprehensive benchmarking, judicious method selection is not currently possible for ANN applications that involve frequent online changes to datasets. Here we address this issue by building upon existing benchmarks for static search problems to provide a new benchmarking framework for big high dimensional dynamic data. We apply our framework to dynamic scenarios modelled after common real world applications. In all cases we are able to identify a suitable recall-runtime tradeoff to improve upon a worst-case exhaustive search. Our framework provides a flexible solution to accelerate future ANN research and enable researchers in other online data-rich domains to find suitable methods for handling their ANN searches.[1]

## 1 INTRODUCTION

Approximate k-Nearest Neighbour (ANN) search is a widely applicable technique for tractably computing local statistics over large datasets of high dimensional discrete samples (Beyer et al., 1999). ANN methods achieve sub-linear search times by trading off search accuracy and runtime. ANN methods are applied in many domains such as image retrieval, robotic localisation, cross-modal search and other semantic searches (Prokhorenkova & Shekhovtsov, 2020). ANN search is well suited to applications where an index structure can be precomputed over a static dataset to then provide a suitable recall-runtime tradeoff.

Achieving an optimal tradeoff for specific dataset properties and application requirements relies on hyperparameter tuning for each suitable ANN method. For instance, graph based indexes can be tuned to achieve high search accuracy while quantisation methods are better suited to perform faster searches with less exact results. In practice, when given a new dataset, there is a significant computational cost to evaluate and select the best performing ANN method. Several ANN benchmarks have been established to guide the selection and parameter tuning required for achieving tractable searches (Aumüller et al., 2017; Matsui, 2020). However, current ANN benchmarks focus on static search problems and cannot be used to inform if any ANN methods are suitable for tackling dynamic search problems where the indexed dataset changes over time.

We observe that current ANN benchmarks do not generalise to dynamic search problems because they perform index construction as an offline process that optimises for search performance on a fixed dataset (Figure 1a). This fails to address the requirements of growing fields such as Machine Learning (ML) where there is a strong need for tractable k-nearest neighbour search on large dynamic sets of high dimensional samples (Prokhorenkova & Shekhovtsov, 2020). For example, local statistics can be extracted by computing neighbourhoods in an embedding space of a learning process, but computing these neighbourhoods requires frequent evaluation of sample locations in the highly dynamic embedding. Due to the lack of suitable ANN benchmarks, achieving tractable

---

[1]Code submitted in supplementary materials and will be available publicly on publication

Figure 1: a) Existing ANN benchmarks evaluate performance using a single batch of searches performed on a static index. b) Our framework generalises to dynamic search problems where batches of index updates occur between batches of searches. Our benchmarks provide an improved model for optimising ANN usage in online data collection and online feature learning.

search performance currently requires extensive evaluation and tuning on the already computationally expensive and highly parameterised systems that could utilise dynamic ANN search. Here, we address this gap and present a novel ANN benchmarking framework for dynamic search problems. Unlike existing benchmarking of ANN search we include the computational costs of constructing and maintaining an index structure throughout a dynamic process.

The main contributions of our work are as follows:

- We present a novel characterisation of the complexity and dynamic variations for ANN search on big high dimensional ($\sim 100$ dimensions) dynamic datasets (Section 2). From this, we generate benchmarks that model domain specific applications in two key categories of dynamic search problems: online data collection and online feature learning (Figure 1b).

- We establish the baseline performance of five promising ANN methods using extended hyperparameter sets to better address the requirements of dynamic search problems (Section 5). We discover that ANN methods such as ScaNN (Guo et al., 2020) and HNSW (Malkov & Yashunin, 2018) can outperform an exhaustive search despite online index overheads.

- We show that our benchmarking framework can successfully identify which ANN method is best suited for a given dynamic search problem. Our framework generates the key tradeoffs for selecting a suitable ANN method and can be extended to additional dynamic search problems and for future ANN research.

## 2 PROPOSED CATEGORISATION OF DYNAMIC SEARCH PROBLEMS

In this section, we identify two key categories of dynamic search problems based on their requirements: online data collection and online feature learning. We also identify key measures that characterise specific instances of both static and dynamic search problems.

ANN search with online data collection or online feature learning requires online index construction (Figure 1b). A major practical advantage of online index construction is that it allows for search information to be fed back in a closed loop fashion. To categorise dynamic search problems, we consider their requirements for online data collection or online feature learning. Autonomous navigation, live internet services and generative learning methods are common examples of online data collection that generate an increasing number of samples over time. An increase in the number of indexed samples will directly increase the computational cost of performing searches. Within many machine learning processes, the training of an embedding space is an example of online feature learning. Updating model parameters during the learning process will update the embedded representation of indexed samples. This update can affect local and global index structures and degrade the performance of subsequent searches.

From a database perspective, we match online data collection and online feature learning with the operations of adding new samples and updating existing samples respectively. Our benchmarks are designed around each of these two operations in order to model the range of dynamic search problems we are interested in. The removal of samples is another fundamental database operation which is often heuristically applied to maintain sample diversity while limiting the total sample count. Removing samples can therefore be viewed as a heuristic that is providing a tractability tradeoff. In this research we omit the remove operator to focus on benchmarking the baseline tractability from ANN methods alone.

In addition to categorising the types of dynamic search problems, we also identify key measures that characterise specific instances of dynamic search problems. Table 1 provides an overview of these measures. Each measure either needs to be specified when defining a benchmark or can be used to evaluate the performance of ANN methods on that benchmark. We identify six measures available to static benchmarking problems. *Sample count* and *sample dimensionality* are enough to specify most static benchmarks and *search accuracy* and *search runtime* are the most common performance evaluation measures. *Initialisation time* and *memory footprint* are less commonly used for performance evaluation (Aumüller et al., 2017; Matsui, 2020).

Table 1: Key measures of high dimensional search problem characteristics that can be used for specifying benchmarks and evaluating the performance of ANN methods. Dynamic search problems require additional benchmark and evaluation measures that are not present in static search problems.

| Measure | Specified by Benchmark | ANN Method Dependant | Static Search Problems | Dynamic Search Problems |
|---|---|---|---|---|
| Sample count | ✓ | ✗ | ✓ | ✓ |
| Sample dimensionality | ✓ | ✗ | ✓ | ✓ |
| Initialisation time | ✗ | ✓ | ✓ | ✓ |
| Memory footprint | ✗ | ✓ | ✓ | ✓ |
| Event type | ✓ | ✓ | ✗ | ✓ |
| Event processing time | ✗ | ✓ | ✗ | ✓ |
| Event frequency | ✓ | ✗ | ✗ | ✓ |
| Event batching | ✓ | ✗ | ✗ | ✓ |
| Search accuracy | ✗ | ✓ | ✓ | ✓ |
| Search runtime | ✗ | ✓ | ✓ | ✓ |
| Search frequency | ✓ | ✗ | ✗ | ✓ |
| Search batching | ✓ | ✗ | ✗ | ✓ |

We identify an additional six measures that apply to dynamic search problems, but not to static ones. To specify a benchmark we include *event type* to select between add events in our online data collection benchmarks and update events in our online feature learning benchmarks. To fully specify the characteristics of a dynamic benchmark we propose the use of frequency and batching measures for both events and searches. *Event batching* and *search batching* define the number of events or searches that will be executed in a continuous block, then followed by a block of the other. *Event frequency* and *search frequency* define how much time is available before another block of events or searches will be available. Lastly, *event processing time* is an additional performance evaluation measure where we capture the runtime cost of online index construction and updates.

The frequency of events and searches has a direct impact on the tradeoff between search accuracy and search runtime. Given a fixed amount of computational resources we have a hard limit on the maximum number of operations that can be performed in any period of time. Examples of these changes include increasing the sensor speed on an autonomous vehicle and increasing the batch gradient decent batch size of a closed loop learning pipeline. To avoid latency, the fixed budget of compute operations must be divided between each of the events and searches processed during the available time period. Increasing the frequency will then result in a reduced amount of compute for processing the events and the searches. For dynamic search problems, we expect that less compute per event will eventually degrade the index quality. Degraded index quality and less compute per search will both ultimately result in lower search accuracy.

The batching of events and searches can also impact the search accuracy-runtime tradeoff in a less direct way. For a fixed amount of compute in a given period of time, we now consider a fixed number of events and searches that will be processed. At each extreme of batching we could have events and searches alternating one after another or we could have a large chunk of events clumped together followed by a large chunk of the searches. Examples here are an autonomous vehicle capturing and querying for each location as it travels and a closed loop learning pipeline completing a full forward pass to then feeding back neighbourhood information about an entire dataset. Theoretically, larger batches of events can provide a total amount of compute that is capable of applying global changes to an index structure, while small batches would be limited to more localised changes. We expect that an index limited to localised changes will degrade in quality, thus reducing search accuracy.

This assertion is supported by the frequent use of dynamic programming and hierarchical structures used in offline index construction for static search problems. Additionally, search batches are always assumed to be mutually independent when benchmarking static search problems. However batching can provide small efficiency gains by increasing the utilisation of parallel hardware. In the context of dynamic search problems we also investigate the performance of ANN methods when consecutive searches do share mutual information.

## 3  FRAMEWORK STRUCTURE AND MODULES

We have designed a flexible, extensible and open access framework to specify and evaluate ANN benchmarks for dynamic search problems. In the previous section we identified twelve important measures for specifying and evaluating dynamic search problems. In this section we detail the three main modules of our framework and how they address these measures. We highlight other novel additions that extend the protocols of existing ANN benchmarks (Aumüller et al., 2018; Matsui, 2020) and discuss the key challenges of benchmarking dynamic search problems. Importantly, we describe each of the frontend components that users of our framework will likely need to interact with as (shown in green in Figure 2).

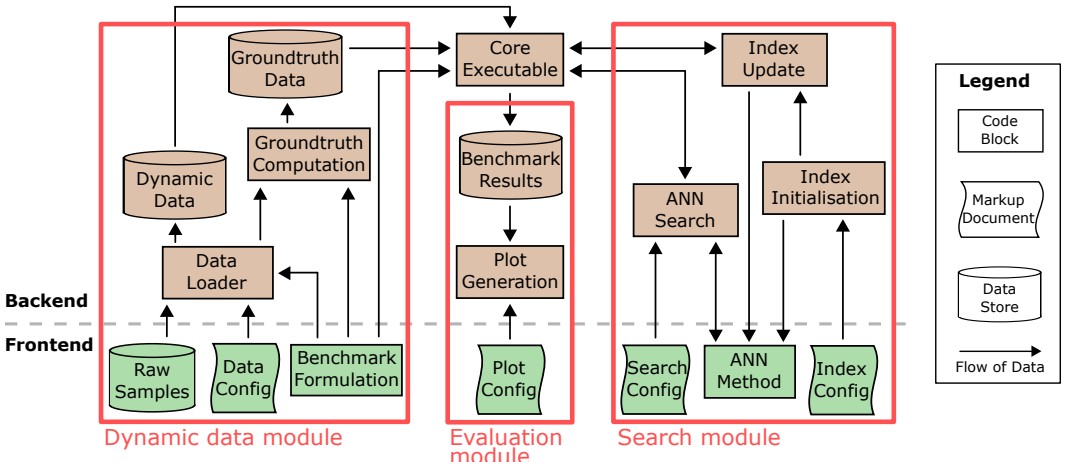

Figure 2: Our framework is built to address the requirements of benchmarking high dimensional dynamic search problems. The three main modules of our framework manage datasets (left), ANN methods (right) and evaluation (middle). Using our framework involves interacting with the frontend components shown in green.

### 3.1  EVALUATION MODULE

Evaluation of dynamic search problems is itself a substantial challenge with each measure on each ANN method at each point in time creating additional outputs that need to be searched and filtered to identify the top performing configurations. A configuration file (*Plot Config*) is used to specify which datasets, ANN methods and performance measures are to be executed. The performance measures listed in Table 1 can be plotted against each other or individual measures can be plotted over time. The time series plots provide a novel perspective that does not exist for static search problems. Each ANN method can be plotted to evaluate its full set of hyperparameter options, or the Pareto frontiers of multiple ANN methods can be plotted to compare their relative performance. Figures in Section 5 and Appendix A showcase some of these plots.

### 3.2  DYNAMIC DATA MODULE

Defining a dynamic search problem in our framework requires three components: a configuration file (*Data Config*), a data blob (*Raw Samples*) and a block of Python code (*Benchmark Formulation*). *Data Config* specifies any hyperparameters for initialising the dataset or augmenting it over time, including the seven benchmark measures from Table 1. *Raw Samples* contains samples for

initialising the dataset and for growing or augmenting it over time, alternatively this can be generated programmatically in the code block. Lastly, *Benchmark Formulation* makes use of helper functions to execute dynamic events, ANN searches and outputting results. In the backend of this module, the groundtruth results, baseline compute times and baseline memory usage are generated by running each newly defined dynamic benchmark using a bruteforce nearest neighbour method. Together this module provides a flexible system for defining and evaluating dynamic benchmarks that are parameterised to model the key properties of specific dynamic search problems.

### 3.3 SEARCH MODULE

Defining a dynamic search problem in our framework requires three components: a search configuration file (*Search Config*), an indexing configuration file (*Index Config*) and a block of Python code (*ANN Method*). The configuration files specify the full hyperparameter ranges for index construction, handling dynamic events and performing ANN searches. The code file provides an interface between the framework and a specific ANN method which includes functions for index construction, handling dynamic events and performing ANN searches. Common ANN methods are currently designed, parameterised and tuned using static search problems. Therefore we expect that the recommended hyperparameter ranges will not align with the optimal range for dynamic search problems. The frequency and batching (of events and searches) is handled within the backend components, this includes the ability to bypass particularly slow event handling to achieve tractable evaluation tradeoffs. Further discussion of hyperparameter tuning and slow event handling is provided in Appendix A.

## 4 APPROXIMATE NEAREST NEIGHBOUR SEARCH METHODS

Approximate k-Nearest Neighbour (ANN) methods are designed to provide favourable tradeoffs between search performance and computational resources (Beyer et al., 1999; Prokhorenkova & Shekhovtsov, 2020). Many viable methods exist and we can differentiate them by the index structures that facilitate the efficient searches. These index structures determine each of the six measurable performance indicators listed in Table 1. Deeper consideration can also be given to underlying factors such as the impact of memory bandwidth and partial index updates for determining the event processing time of a method. In this section we provide an overview of the methods we have evaluated. Our selection criteria is to focus on methods with an optimised, publicly available implementation that can be interfaced with Python code. Table 2 summarises the type of index structure used by each method, if it is an exact search, whether samples can be added without rebuilding the index (incremental construction for online data collection) and whether sample values can be updated without rebuilding the index (for online feature learning). Methods that do not support these functions will have additional construction overheads as the entire index is rebuilt each time a dynamic event is processed. Rebuilding an index will often result in large *event processing time* that is then partially mitigated by the slow event handling discussed in Appendix A.

Table 2: ANN methods in current and common usage for static search problems. We extend and evaluate each method with our dynamic search problem benchmarks.

| Index Type | ANN Method | Exact Search | Incremental Construction | Update Samples |
|---|---|:---:|:---:|:---:|
| None | Bruteforce | ✓ | - | - |
| Quantisation | IVFPQ (Johnson et al., 2019) | ✗ | ✓ | ✗ |
| | ScaNN (Guo et al., 2020) | ✗ | ✗ | ✗ |
| Tree | Annoy (Bernhardsson, 2013) | ✗ | ✗ | ✗ |
| | k-d Tree (Pedregosa et al., 2011) | ✗ | ✗ | ✗ |
| Graph | HNSW (Malkov & Yashunin, 2018) | ✗ | ✓ | ✓ |

The baseline approach to nearest neighbour search exhaustively computes distances between each query and each sample in a dataset while maintaining a set of the k-nearest samples for each query. This bruteforce method can be accelerated with parallel computation of partial distances, full distances and simultaneous queries. A linear recall-runtime tradeoff can be achieved by only computing

distances to a subset of samples (reported as Baseline in Section 5). Importantly, there are no additional overheads for constructing or maintaining an index over the samples. Currently there is no consensus on which ANN methods, if any, are most suitable to replace bruteforce search in dynamic search problems. Recent papers include examples of using bruteforce search (Komorowski, 2021; Vidanapathirana et al., 2021), k-d trees (Kim & Kim, 2018; Xu et al., 2021) and graphs (An et al., 2019; Schubert et al., 2021) to address similar dynamic search problems.

Quantisation methods (Johnson et al., 2019; Guo et al., 2020) cluster a dataset into local neighbourhoods where efficient residual distances are used. Quantisation methods are related to hashing methods with their regular partition of a high dimensional space. However, quantisation methods achieve this partition by grouping samples around cluster centers rather than cutting up the space with hyperplanes. Clustered samples are then mapped to a codebook using a reduced dimensional space within each cluster. Cluster centers are learned from the intrinsic structure of a dataset and this provides a significant improvement to recall performance, but at the expense of a larger index construction time.

Tree methods (Bernhardsson, 2013; Bentley, 1975; Pedregosa et al., 2011) connect dataset samples into a traversable index structure with a single source and no cycles. Tree methods are frequently used for their fast index construction and searches. Compared to quantisation methods, tree methods have an additional memory cost of storing the edge information that enables the search paths. Traversing a tree involves many local decisions that utilise an increasingly smaller subset of the global information as sample dimensionality increases. As such, higher recall can only be achieved with a significant amount of backtracking on a single tree or by searching in parallel over a forest of semi-redundant trees. Despite these limitations, standard k-d tree implementations continue to be applied to high dimensional search problems (Kim & Kim, 2018; Xu et al., 2021).

Graph methods (Arya & Mount, 1993; Harwood & Drummond, 2016; Malkov & Yashunin, 2018) generalise tree methods by allowing any set of directed edges that connect all dataset samples. Graph methods are some of the most recent and promising methods due to favourable recall-runtime tradeoffs in existing ANN benchmarks. Like tree methods they require the additional storage of edge information. Graph methods are better suited for indexing big high dimensional data due to their lack of an explicit hierarchical structure which ensures that all local regions are well connected to other areas of the graph. However, the selection of an appropriate set of edges is computationally intensive, leading to large index construction times. Hierarchical navigable small world graphs (HNSW) (Malkov & Yashunin, 2018) avoid this cost by semi-randomly selecting a set of edges with desirable statistical properties.

## 5 BENCHMARKING EXPERIMENTS

We evaluate the ANN methods of interest across our proposed benchmarks and report the key findings. We present the Pareto frontier of recall-runtime performance for each method. Runtime results are reported on a logarithmic scale as a ratio against the full bruteforce runtime (*speedup over bruteforce* combines total *event processing time* and *search runtime*). We limit all methods to run on a single AMD EPYC 7543 CPU thread (3.7 GHz turbo, 256MB cache) with up to 10GB RAM. This firstly aims to provide results that are more analogous to theoretical complexity analysis. Comparing the theoretical complexity of different ANN algorithms is otherwise impractical due to algorithm performance being dependant on dataset properties. And secondly, a single thread mitigates performance biases that can occur at particular parallelisation thresholds. However, algorithms are not restricted from performing thread level parallelisation by taking advantage of hardware specific SIMD instructions for I/O and arithmetic. Recall results are reported as an average, where each search returns the top 50 nearest neighbours for a query and is scored against the ground truth top 50. All neighbours are weighted equally for scoring. This choice of neighbourhood size and weighting is selected to be realistic for the real-world applications being modelled. In practice, these moderately large neighbourhood sets are sized to accommodate inherent noisiness in the samples and a secondary metric would be used to re-weight these shortlisted samples for downstream usage.

Within our benchmarking framework we have implemented two datasets targeted towards broad application areas. Firstly, our online data collection dataset is modelled around real-time applications where each subsequent query is appended as a new sample to the dataset. We use various subsets of the 128 dimensional sift1M datasetJegou et al. (2008), with *add events* growing each dataset to

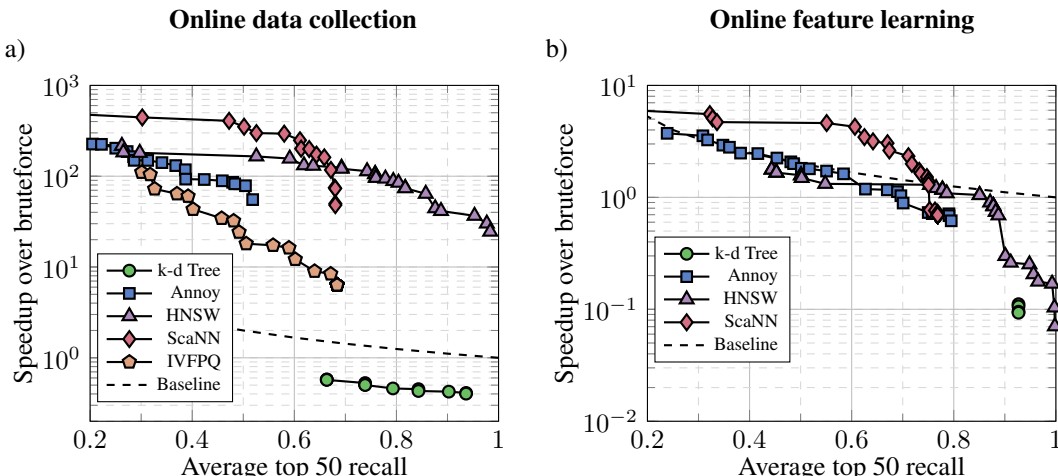

Figure 3: Recall vs runtime performance against a bruteforce baseline using our extended hyper-parameters and tuning for dynamic search problems. a) An online data collection benchmark with 100k samples, 128 dimensions, 100k added events, event and search batch size 1. b) An online feature learning benchmark with 5k samples, 96 dimensions, 100k update events, event and search batch size 200.

double its initial size and an additional hyperparameter to adjust mutual information by interpolating consecutive queries. This allow us to explore applications where either each query is assumed to be independent of any other, or where consecutive queries are assumed to have some degree of commonality. Initial samples are expected to be representative of the total sample pool, altering this is expected to degrade the performance of some methods. Secondly, our online feature learning dataset provides a generalised but flexible model for machine learning applications. It applies *update events* to model a dataset where samples are converging across incremental updates. We use various subsets of the 96 dimensional deep1B dataset Babenko & Lempitsky (2016), with additional hyperparameters to simulate training epochs, batch size, convergence rate, a clustering coefficient and if groundtruth is measured after each batch or each epoch. This allows users to explore various machine learning applications and can be further extended to suit more specific goals.

Our results in Figure 3 show that existing ANN methods can achieve useful performance by exceeding a bruteforce baseline on our two applications. For the online data collection benchmark in Figure 3a, k-d tree (green circle) is the only method that does not outperform the bruteforce baseline. This validates that k-d trees are not suitable in these dynamic high dimensional search problems, but also that other methods are indeed capable of providing a desirable recall-runtime tradeoff. HNSW (Malkov & Yashunin, 2018) (purple triangle) is the only algorithm outperforming bruteforce above 70% recall, while ScaNN (Guo et al., 2020) (red diamond) is the fastest algorithm below 70% recall. The benchmarks produced by our framework succeed in identifying crossover points, such as this one between HNSW and ScaNN, which are critical knowledge when selecting an ANN method for a given application. Comparing to the online feature learning benchmark in Figure 3b, all methods perform considerably worse compared to the baseline. This is in part due to the *update events* (discussed in Section 2) affecting all samples in an index, while the *add events* have a more local impact. For the online feature learning benchmark, no method is faster than bruteforce beyond 78% recall, or more than $5\times$ faster beyond 20%. K-d trees perform well below the baseline and we are unable to produce results for the current IVFPQ implementation. Further discussion on time series performance, hyperparameter tuning and dataset scale is provided in the Appendix A.

We verify the importance of the novel characteristics defined in Table 1 and Section 2 by varying the benchmark specified hyperparameters that have been added for dynamic search problems. Specifically these hyperparameters are the frequency and batching of both events and searches. The impact of each hyperparameter on the ANN search performance indicates if that parameter is an important consideration when creating benchmarks for real-world dynamic search problems. In Figure 4a we see that increasing the event frequency increases the difficulty of the dynamic search problem. The

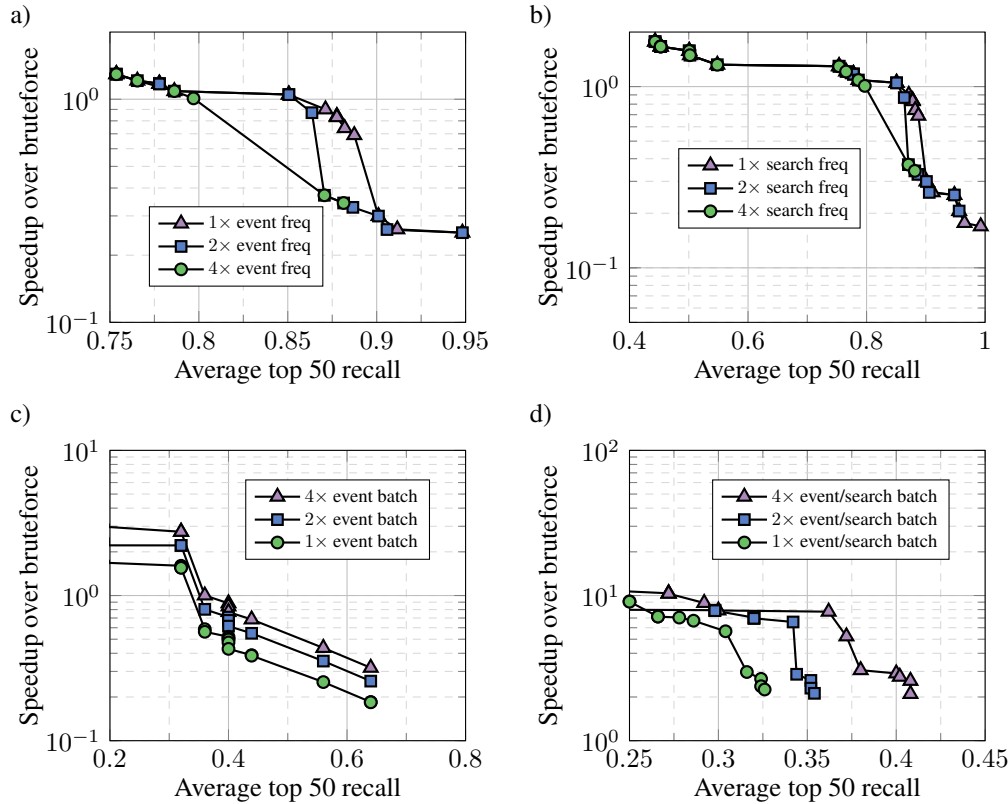

Figure 4: Recall vs runtime performance when varying the frequency and batching of both events and searches. Shown for an online feature learning benchmark with 5k samples, 96 dimensions, 100k update events, event and search batch size 200, HNSW method (Malkov & Yashunin, 2018). a) Increasing event frequency with fixed search frequency. b) Fixed event frequency with increasing search frequency. c) Increasing event batching with fixed search batch size. d) Increasing event and search batch size together.

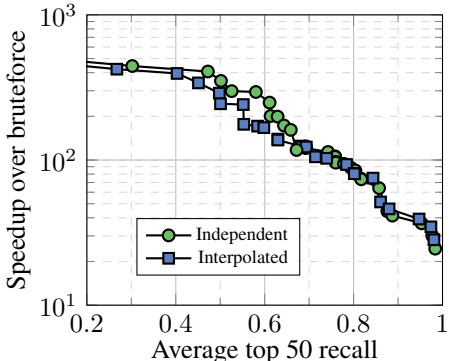

Figure 5: Recall vs runtime performance for temporally dependant events. Shown for an online data collection benchmark with 100k samples, 128 dimensions, 100k added events, event and search batch size 1, HNSW method (Malkov & Yashunin, 2018), 10% interpolation.

recall-runtime tradeoff moves towards the origin until it saturates. Figure 4b demonstrates similar behaviour for increases in search frequency. Figure 4c and Figure 4d shows that significant performance differences can also be induced by varying the event batching. Collectively from the results in Figure 4, it is evident that each of the benchmark hyperparameters we have defined are important

considerations when characterising a dynamic search problem for benchmarking. Further discussion on the relationship to time series performance is provided in the Appendix A.

Lastly, we consider the role of mutual information that exists between consecutive search batches due to temporal consistency in dynamic search problems. In the context of ANN search, this temporal consistency creates higher overlap between groundtruth neighbourhood sets in consecutive search batches. We model temporally consistent events by augmenting our datasets to interpolate between samples over time. In Figure 5 we plot the ANN search performance of the interpolated dataset against performance of the non-interpolated dataset. From our results we conclude that the ANN methods we evaluate fail to exploit mutual information between consecutive search batches. These methods do not attempt to update their internal parameters or search strategies based on recent queries. Ideally, an ANN method designed specifically for dynamic search problems would utilise information from recent queries to generate considerably improved recall-runtime performance for the interpolated dataset.

## 6 CONCLUSIONS AND FUTURE WORKS

We present a novel benchmarking framework to address the significant gap in utilising ANN methods for dynamic search problem. ANN methods are seeing increasing use in domains such as machine learning where there is a need to distill local statistics from large amount of high dimensional data. However the dynamic nature of this data is at odds with the static search problems that are used when designing and benchmarking current ANN methods. Our framework directly addresses this issue by allowing us to define new benchmarks that give a first indication of how current ANN methods will actually perform in dynamic contexts.

Our generalised approach to dynamic search problems has enabled us to categorise and implement benchmarks that model a number of real-world problems. Specifically we focus on the dynamic search problems that feature online data collection and online feature learning. Within these two categories we have also explored the effects of dynamic search problem characteristics and temporal dependence between consecutive samples. Our extensible framework allows for future work to add new categories of dynamic search problems, build upon the current ones or replay real-world data captures. In particular, our work does not currently extend to include the fundamental database operation of removing samples. Removing samples becomes an important consideration when the pruning of old or low saliency samples is needed for bounding memory consumption.

Our benchmarks show that judicious selection of ANN methods is critical for efficient performance on dynamic search problems. We find that a standard implementation of k-d trees performs significantly worse than an exhaustive search. This is despite continued usage in dynamic high dimensional data contexts (Kim & Kim, 2018; Xu et al., 2021). In contrast, HNSW (Malkov & Yashunin, 2018) and ScaNN (Guo et al., 2020) perform very favourably on our dynamic search problems. Again the extensibility of our framework supports the inclusion of additional ANN methods as well as the evaluation of new or altered methods. We hope the performance benchmarks established in this paper will be a launchpad that is quickly exceeded by further research.

We have demonstrated that additional hyperparameters and hyperparameter tuning can adapt existing ANN methods from static to dynamic search problems. However, there are still large research gaps in designing ANN methods that are specifically tailored for exploiting the full information available in dynamic search problems. ANN methods with slow index construction benefit greatly from reducing the frequency of event and batching them into larger index updates. Some ANN methods have been designed to exploit mutual information from application specific temporal relationships (Garg & Milford, 2021). Extending general ANN methods in this way could provide a large performance gain on these dynamic search problems.

Dynamic search problems are not going away, they are instead becoming more wide spread due to increasing demands for fast and realtime processing of big and rich data streams. This means there is lots to be gained from a deep and thorough exploration of which dynamic search problems are common place, and which ANN methods are best suited to address their specific demands. Our benchmarking framework is well positioned as a launching point for others who are interested in exploring and contributing to the many open problems that exist in this area.

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

## A  APPENDIX

Each datapoint from the figures in Section 5 is an average over a series of events and searches that define that benchmark as a particular dynamic search problem. We can further analyse the performance of each ANN algorithm by considering the time series of how performance measures (as defines in Table 1) vary throughout benchmarking. Figure 6 illustrates these temporal variations for the *search performance* measure. We compare each ANN method using the hyperparameter set identified at the harmonic mean of that methods Pareto frontier. For analysing and optimising individual ANN methods, time series results across hyperparameter sweeps of that method can be compared instead.

In Figure 6 we see that recall performance of all methods is relatively consistent across this benchmark. Because this is an online data collection benchmark we might expect to see a decrease in recall as the dataset grows from 100k to 200k samples, and similarly, from degradation as the index is continuously expanded and rebuilt. However, Annoy is the only method with a slight downwards trend in its recall performance. For all other methods, the recall remains flat over time and we would need to consider the time series of other performance measures to confirm if the cost of processing a larger dataset is being paid elsewhere. For both the k-d Tree and ScaNN methods we can see periodic oscillations in the recall. These oscillations are caused by the *slow event handling* in the backend of our *search module* (introduced in Section 3). Reducing the regularity of index updates is important for reducing average runtime, however it results in decreasing recall until the next index update is performed. In practice, we find that slow searches create downstream latency for systems using the outputted ANN results, while slow event handling can instead be absorbed by the recall-runtime tradeoff. As such, maintaining a consistent *search frequency* is more important than prioritising event handling. Our framework incorporates this behaviour by extending all ANN methods with an optional hyperparameter that delays index updates for a specified number of events. This delay artificially lowers the *event frequency* and increases the *event batching* (as discussed in Section 2 and Figure 4).

Figure 7 presents additional plots to illustrate the role of hyperparameter tuning and *sample count*, on the two dynamic search problems we have analysed. Comparing Figure 7a to 7b and Figure 7d to 7e shows the importance of additional hyperparameters and hyperparameter tuning for dynamic search problems. Figures 7a and 7d contain results when using recommended hyperparameter sets that have been tuned on static search problems. Whereas, Figures 7b and 7e replicate the dynamic search problem results from Figure 3. Our tailored approach to hyperparameter tuning results in improved performance for all ANN methods. This comparison highlights the critical importance of our benchmarking framework - the best practice approaches for static search problems do not translate directly to dynamic search problems. To address the requirements of dynamic search problems we extend the hyperparameter ranges of each method in directions that result in higher computational efficiency. This efficiency will typically be gained at the expense of recall. Additionally, we add an additional hyperparameter for optional management of *slow event handling*. Again we find that a reduction in recall is typical when the slow event handling is applied.

Lastly, in Figures 7c and 7f we illustrate the relationship between sample count and runtime, on log-log plots for the two dynamic search problems we have analysed. For each ANN method,

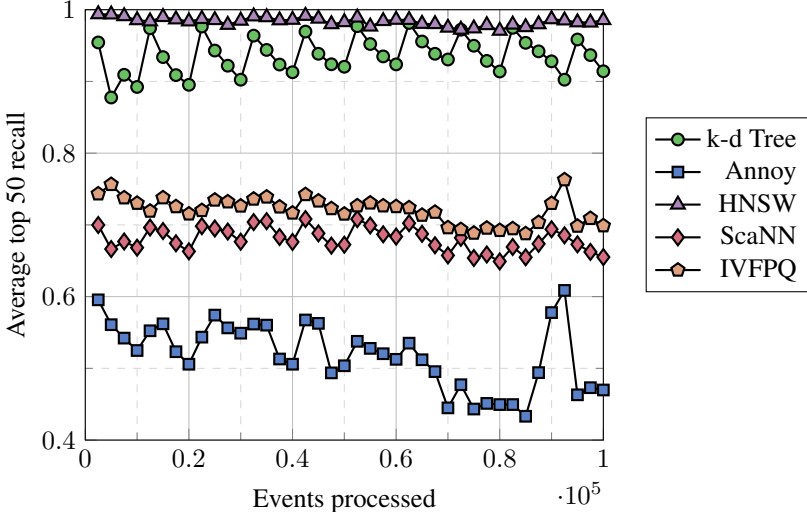

Figure 6: Events processed vs recall performance for the harmonic mean of the Pareto frontiers. Shown for an online data collection benchmark with 100k samples, 128 dimensions, 100k added events, event and search batch size 1.

we plot the best runtime performance that achieves at least $50\%$ recall for a dataset of a given sample count. At $50\%$ recall the baseline bruteforce method can choose to ignore half of the dataset, so the baseline cutoff is shown at 2. Here we see a trend for the dynamic search problems that is consistent with what has previously been found for static search problems. This trend shows that ANN methods increasingly improve their performance over a baseline bruteforce search as sample count increases. Extending our prior analysis of Figure 3, we identify the crossing points for the online feature learning benchmark where each ANN method becomes more efficient than the baseline. These crossing points are of similar interest to the crossing points of ANN methods with each other, as this information is needed to guide users towards the most suitable ANN method for a given application. Additionally by comparing Figures 7c and 7f we verify that *update events* (as introduced in Section 2) do result in worse performance than the *add events*, even when sample count is taken into consideration. For the same sample count and ANN method, we consistently see lower performance on the online feature learning benchmarks than the online data collection benchmarks. Further analysis of each individual ANN method will be required to identify the root cause of this performance difference between *event types*, and potentially to then further improve ANN search performance for online feature learning.

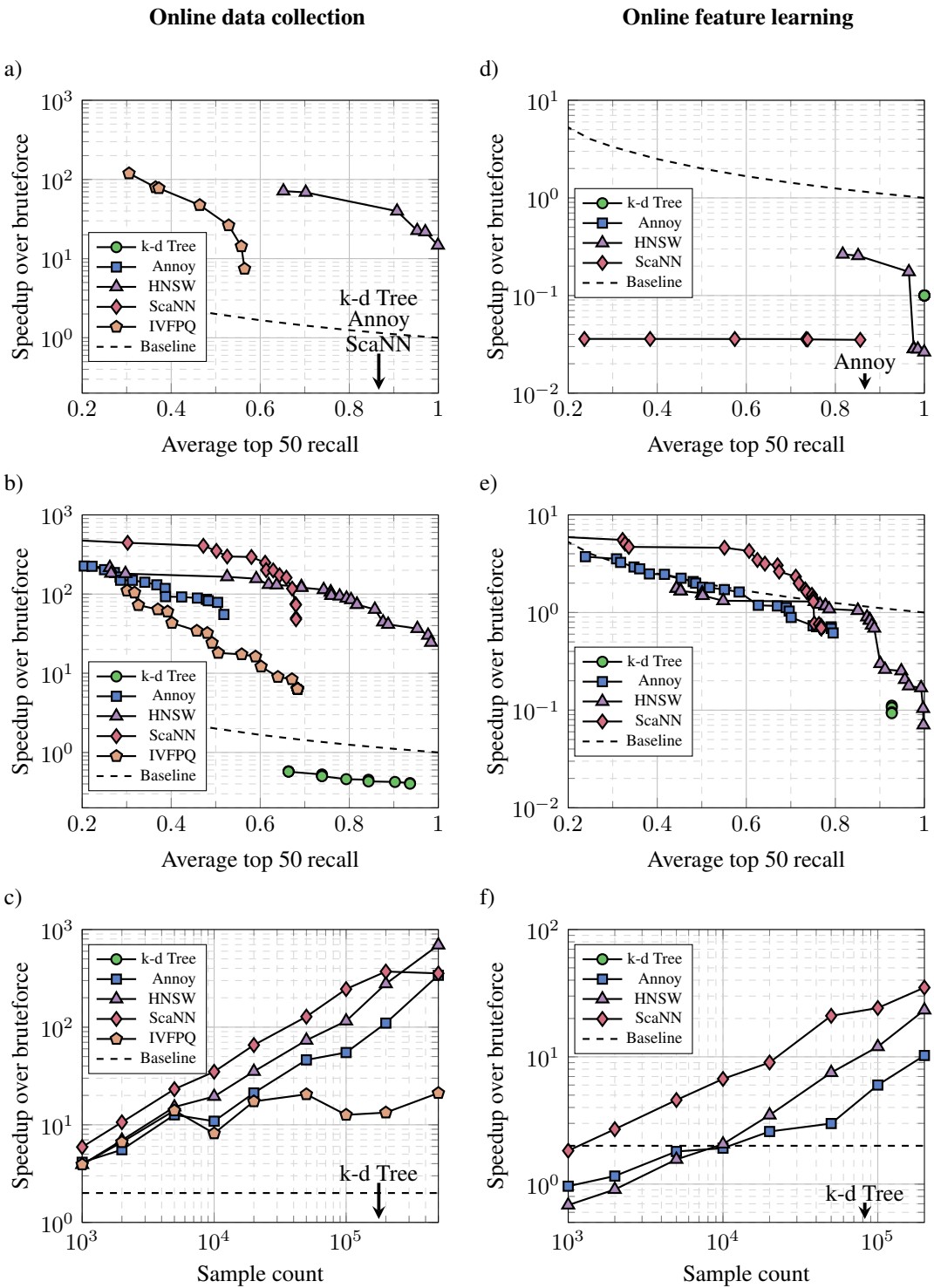

Figure 7: Recall performance against a bruteforce baseline. a, b, c) An online data collection benchmark with 100k samples, 128 dimensions, 100k added events, event and search batch size 1. d, e, f) An online feature learning benchmark with 5k samples, 96 dimensions, 100k update events, event and search batch size 200. Top - Recall vs runtime with hyperparameter tuning taken from static search problems. Middle - Recall vs runtime with our extended hyperparameters and tuning for dynamic search problems. Bottom - Runtime vs sample count with datasets ranging from 10k to 500k samples and events evaluated at 50% recall.

