# OpenReview forum: "Benchmarking Approximate k-Nearest Neighbour Search for Big High Dimensional Dynamic Data"
_ICLR.cc/2023/Conference — Submitted to ICLR 2023_

### Official Review · Reviewer_brtU · 2022-10-24

**Confidence:** 5
**Correctness:** 3
**Technical Novelty And Significance:** 2
**Empirical Novelty And Significance:** 2
**Recommendation:** 3

**Clarity, Quality, Novelty And Reproducibility:**

Suggestions for improving paper:

**Clarity**
1. The settings "online feature learning" and "online data collection" need to be explained more clearly. What does it mean for the ANN index? Presumably "online data collection" just means that new points are inserted into the index? Does online feature learning mean that the underlying model is being trained and this
2. More clarity needed on temporally dependent events -- what are the sources of these data and how are the points correlated?
3. IVFPQ and SCANN: Is the quantization changed over time? How is the initial quantization computed when zero points are provided? Does the initial quantization generalize over time? Same question for clustering in IVF.


**Quality**
1. The main parameter varied in Fig 4 is inserts:search ratio. Isn't it clear that dynamic ANN indices can outperform bruteforce? I feel there are other experiments that merit more attention than this plot. See the remaining points:
2. What is the total pool size of points from which the experiments sample data? You could use the 1B scale datasets released at big-ann-benchmarks.com and explore various permutations of insertions and resultant effects.
3. Lack of parallelism. Support for parallelism is an important feature of algorithms and should be measured. In practice, a single dynamic index would be deployed on a multicore machine and could be supporting changes and searches at the same time.
4. The conclusion could be more specific and focused on the learnings from the experiments as opposed to general comments on the ANN space.
5. It unclear that FAISS IVFPQ as used in this paper has been tuned well. There are recent discussions on how FAISS IVFPQ can compete with SCANN: See https://medium.com/@kumon/similarity-search-scann-and-4-bit-pq-ab98766b32bd and https://github.com/facebookresearch/faiss/wiki/Fast-accumulation-of-PQ-and-AQ-codes-(FastScan)
6. Please consider taking the best 2 or 3 algos for each dataset in ann-benchmarks.com to ground the choice of algorithms you benchmark.

**Novelty**
1. The authors don't propose novel algorithms but describe an evaluation framework, which is OK. Given the lack of evaluation for dynamic ANN, this work is a good first step.

**Reproducibility**
1. Consider organizing the paper and code around (A) APIs a new algo must support to  be measured in your framework, and (b) use cases and datasets that that the eval framework considers important. These eval code should describe and support these two orthogonally.
2. What are the datasets used in this paper and their source?

**Strength And Weaknesses:**

Strengths:
1. Authors list various parameters of importance to dynamic setting.
2. Authors identify the lack of comparison for dynamic indices as a major missing piece on ANN literature. The research is timely.
3. Authors identify that its possible to get substantial speed up over

Weaknesses:
1. The authors miss an opportunity to present clear and comprehensive experiments. A few examples of points important for dynamic indices but missed here:
- How do batch-build vs streaming-build indices compare in quality?
- No measurement of indexing effort vs search quality. There is a direct trade off between index construction complexity and "search complexity vs recall", the latter being the property studied in all plots. This is especially so for graph indices. A study of dynamic indices for graph indices could measure this trade off?
2. Delete operation is ignored. This is important in practice, as a lack of delete operation forces a (expensive) rebuil of the indexafter substantial modification. While most indices measured in this paper do not support delete operation, there is recent work on deletes that this paper does not cite.  These papers also include algorithms at least as good as the ones studied in the paper for inserts.
  - Proximity Graph Maintenance for Fast Online Nearest Neighbor Search [Zhaozhuo Xu, Weijie Zhao, Shulong Tan, Zhixin Zhou, Ping Li]
  - FreshDiskANN: A Fast and Accurate Graph-Based ANN Index for Streaming Similarity Search [Aditi Singh, Suhas Jayaram Subramanya, Ravishankar Krishnaswamy, Harsha Vardhan Simhadri]
  - Parallel Nearest Neighbors in Low Dimensions with Batch Updates [Magdalen Dobson, Guy Blelloch]
3. Do these indices generalize as dataset changes or increases in size? How do we quantify this? I am especially concerned about the quantization and clustering schemes and how they are bootstrapped and maintained over time. These details are missing in the paper.




**Summary Of The Paper:**

The paper proposes an evaluation framework for approximate nearest nearest neighbor search algorithms in a dynamic setting (index set changes over time). It measures various algorithms in this space and concludes that is possible to tune the algorithms so that they are far better than bruteforce search for high dimensional data.

**Summary Of The Review:**

While a greatly welcome first step in the direction of benchmarking dynamic ANNS indices, the work falls short on several fronts -- metrics, evaluation set up, choice of operations, quality of tuning, description and coverage of use cases. Therefore, it does not rise to the quality necessary to be a useful reference in this area.

---

> ### Author Response · Authors · 2022-11-14
> **Response to reviewer comments**
>
> Thank-you for the time and consideration put towards this review, we are pleased to hear that the utility and timeliness of our work has been well conveyed. Regarding your other comments:
>
> W1a: Future work. We agree that comparisons to streaming methods are an interesting future direction to consider, but are currently outside the scope of this work. In this work we focus towards addressing open questions in how to categorise dynamic search problems and measure index quality over time.
>
> W1b: Clarification. Currently our framework measures both 'Event processing time' (indexing effort) and 'Search runtime' (search complexity), with our 'Speedup over bruteforce' plots providing the sum of these values. This clarification will be added to the manuscript. Our framework does also include code to output the non-aggregated runtime results. More generally it covers the trade-offs between each pair of measures, which can show comparisons between ANN methods and between hyperparameter tunings of a single ANN method. Currently they are not included in this paper due to the scale of the data and their sparse utility for a general audience. Are there specific trade-offs that you believe add value for the ICLR community?
>
> W2a: Future work. We agree that remove events are an interesting future direction to consider, but are currently outside the scope of this work. For the purposes of stress testing ANN performance, add and update events increase problem complexity and so are of higher importance than remove events which aim to reduce complexity.
>
> W2b: Future work. Although we agree that the inclusion of additional methods will continue to increase the value of this work, the inclusion of saliency heuristics is currently outside the scope of this work. We intend that the release of our framework will encourage others to contribute specialised methods and datasets.
>
> W3a: Clarification. Our framework does include code to output these results. Specifically plotting a performance measure over time for different hyperparameter tunings of a single ANN method. While not a direct measure of generalisation, these measures do adequately proxy for it. Currently they are not included in this paper due to their sparse utility for a general audience. Are there specific use cases that you believe add value for the ICLR community?
>
> W3b: Clarification. Index maintenance is briefly described at the end of Sec 4 paragraph 1 and in Table 2. We follow the cited authors recommendations of updating an index, otherwise a larger event processing cost is paid due to the index being rebuilt. Our framework smooths these costs using the 'slow event handling' discussed in Appendix A paragraph 2. This clarification will be added to the manuscript.
>
> C1: Clarification. Online data collection is implemented as you describe. Our online feature learning dataset provides a generalised but flexible model for ML/DL applications. It models a dataset that is converging through incremental updates. We use subsets of the 96 dimensional deep1B dataset (Efficient indexing of billion-scale datasets of deep descriptors, Babenko el al, CVPR, p2055-2063, 2016) with additional hyperparameters to simulate training epochs, batch size, convergence rate, a clustering coefficient and if groundtruth is measured after each batch or each epoch. This allows users to explore various ML/DL applications and can be further extended to suit more specific goals. This clarification will be added to the manuscript.
>
> C2: Clarification. The online data collection dataset has been implemented with a hyperparameter to simulate an amount of shared information between consecutive queries. The results in figure 5 illustrating performance between 0% and 10% interpolated datasets. This clarification will be added to the manuscript.
>
> C3: Clarification. The number of samples given for each dataset in the paper is the initial number of samples that are available for training. Each add event is then increasing the total number of samples being indexed. If the initial samples are not representative of the total sample pool then these methods will generalise poorly. This clarification will be added to the manuscript.

---

> > ### Author Response · Authors · 2022-11-14
> > **Continued**
> >
> > Q1: Clarification. See W1b above.
> >
> > Q2: Clarification. See C1 above.
> >
> > Q3: Future work. We agree that the inclusion of parallelisation will continue to increase the value of this work, however it is currently outside the scope of this work. As discussed in Sec 5 paragraph 1 we believe single thread evaluation is the benchmarking strategy with the most general utility, but no single strategy can fully control for the 'hardware lottery'. We intend that the release of our framework will encourage others to contribute specialised hardware specific benchmarks.
> >
> > Q4: Clarification. As mentioned in W1b and W3a above, there is an unwieldy number of comparisons that can be generated from this framework and other specific results are not included in this paper due to their sparse utility for a general audience. This would only grow with the potential inclusion of additional indexing methods, streaming methods, remove events and evaluations on parallel hardware. Are there specific results that you believe add the most value for the ML and ANN communities?
> >
> > Q5: Clarification. The configuration of our parameters sweeps is supplied in the supplementary materials under \url{./dyann/conf/algo/}. We believe that the FAISS IVFPQ implementation has been heavily engineered toward optimal performance on parallel hardware and is likely loosing that advantage once all algorithms are able to fully utilise the computational resources in a single thread paradigm.
> >
> > Q6: Future work. We agree that the inclusion of NGT, N2, pynndescent and vamana will continue to increase the value of this work, however it is currently outside the scope of this work. We intend that the release of our framework will encourage others to contribute specialised hardware specific benchmarks.
> >
> > R1: Accepted as comment. Thank-you for the suggestion. Can you recommend any publications or publication venues that fit this style?
> >
> > R2: Accepted. Thank-you for highlighting the missing implementation details, we will clarify this in the manuscript. From the code documentation in the supplementary material - Online feature learning source \url{https://github.com/matsui528/deep1b_gt#bonus-deep1m} under MIT license and Online data collection source \url{http://corpus-texmex.irisa.fr/} under CC0 1.0 license.

---

### Official Review · Reviewer_vEWF · 2022-10-25

**Confidence:** 4
**Correctness:** 4
**Technical Novelty And Significance:** 3
**Empirical Novelty And Significance:** 3
**Recommendation:** 6

**Clarity, Quality, Novelty And Reproducibility:**

The paper is written well and clear. The efforts made in this direction of benchmarking is needed. Code is provided.

**Strength And Weaknesses:**

Strengths:
1) The paper addressed the much-needed benchmark space for the ANN algorithms.
2) The benchmarks are provided for the additional metrics for dynamic search, i.e., Event type, event processing time, event frequency, event batching, Search frequency, and search batching.

Comments:
1) I might have missed it, but the details about the datasets need to be clarified.
2) More ANN algorithms: The authors have done a great job including many well-known ANN algorithms in the benchmark. They have included Quantization based, Tree-based, and Graph-based algorithms. Including pure partitioning-based methods like IVFFLAT, LSH table, or learned indices like Neural LSH (Dong et al., 2019) and BLISS (Gupta et al., 2022) will be great.
3) Are there any similar benchmarks for dynamic search problems?
4) It will be great to have a table like Table 1 with different methods listed with their performance against the given measures. The performances can be depicted by ranks or markers like ++ , + , - - etc.

*Learning sublinear-time indexing for nearest neighbor search. arXiv preprint, arXiv:1901.08544, 2019. URL http://arxiv.org/ abs/1901.08544.

*A billion scale index using iterative re-partitioning. In Proceedings of the 28th ACM SIGKDD Conference on Knowledge Discovery and Data Mining, KDD ’22, pp. 486495, 2022.



**Summary Of The Paper:**

The paper identifies the key measures of high-dimensional search problems and utilizes them to benchmark the ANN methods for dynamic search. The paper benchmarks state-of-the-art ANN algorithms - HNSW, SCANN, k-d Tree, IVFPQ, and Annoy on speedup against the brute-force near neighbor. The online data collection (adding items online) and online feature learning (updating item features online) aspects are covered for the dynamic search.

**Summary Of The Review:**

The paper is written well and addresses the need for benchmarking on dynamic search. Some additional experiments and clarity will make it an excellent paper.

---

> ### Author Response · Authors · 2022-11-14
> **Response to reviewer comments**
>
> Thank-you for the time and consideration put towards this review, we are pleased to hear that the utility of our work has been well conveyed. Regarding your other comments:
>
> C1: Accepted. Thank-you for highlighting the missing implementation details, we will clarify this in the manuscript. From the code documentation in the supplementary material - Online feature learning source \url{https://github.com/matsui528/deep1b_gt#bonus-deep1m} under MIT license and Online data collection source \url{http://corpus-texmex.irisa.fr/} under CC0 1.0 license. Other values are provided in the figure captions, but can be summarised together for clarity.
>
> C2: Future work. Although we agree that the inclusion of additional methods will continue to increase the value of this work, the inclusion of a pure partitioning method is currently outside the scope of this work. In general the partitioning of high dimensional space gives poor results for ANN search without the use of specialised heuristics. We intend that the release of our framework will encourage others to contribute specialised methods and datasets.
>
> C3: Clarification. To the best of our knowledge, ours are the first benchmarks for dynamic ANN.
>
> C4: Clarification. Our framework does include code to output these results needed for this kind of table. However a table of this kind may have limited value for a general audience. When each table cell associates an ANN method and a performance measure then at least one other performance measures needs to be fixed to provide useful results. For instance if an application requires a maximum memory footprint and minimum recall are maintained, then a table can be generated by optimising over each methods hyperparameters and reporting the best and worst performers in terms of various runtime measures.

---

### Official Review · Reviewer_GRS7 · 2022-10-25

**Confidence:** 4
**Correctness:** 3
**Technical Novelty And Significance:** 2
**Empirical Novelty And Significance:** 3
**Recommendation:** 5

**Clarity, Quality, Novelty And Reproducibility:**

Overall, this is a well-written paper with a good framework that looks well-motivated. However, considering this paper's strengths and weaknesses, the quality and novelty do not seem very strong to me.

For the reproducibility concern, as they declare they will make the code available, the reproducibility should be somewhat assured.

**Details Of Ethics Concerns:**

No.

**Strength And Weaknesses:**

Strengths:

1. The motivation is clear. Even though there exist a few benchmarking frameworks for ANN search, they are designed for static search problem that assumes the dataset is fixed. It seems that this work is the first to consider the dynamic search problem.

2. The authors identify many measures for the dynamic search problem and use experiments to confirm that these measures are valid for the benchmarking framework they proposed.

3. The paper is well-written and easy to read.

Weaknesses:

1. My primary concern is that the technical depth seems somehow limited. I appreciate the six new measures they proposed and the experiments for the five ANN methods they conducted, but most of the measures are straightforward, and all five ANN methods are open-sourced. The framework will be more promising if they can build a cost/analytical model to analyze the relationships of different measures.

2. In the 2nd paragraph of Section 2, they provide many examples of online data collection and online feature collection. It would be more convincing if they could add references for support.

3. In the experiments, what are the datasets they used for online data collection and online feature learning? They provide the dataset statistics, but I cannot find what datasets they used.

4. The experiments only report the curves of speedup and recall. Can they also report the results for other measures, such as the event processing time, initialization time, and memory footprint?

5. For the last experiment (Figure 5), can you explain why the five ANN methods fail to exploit mutual information between consecutive search batches?

**Summary Of The Paper:**

This paper looks at the Approximate Nearest Neighbor (ANN) search problem for high-dimensional dynamic data. The authors propose a new benchmarking framework to fill the gap in selecting the satisfied ANN method for dynamic ANN search. They focus on two requirements: online data collection and online feature learning, and identify six additional measures for the dynamic search problem. Finally, they evaluate five ANN methods and show that the proposed benchmarking framework can determine which ANN method is most satisfactory for a given dynamic search problem.

**Summary Of The Review:**

In summary, I think this paper is well-written and has a clear motivation, but currently, I feel the technical depth is limited, and I have some concerns about the experiments. Thus, I initially provided a borderline reject.

---

> ### Author Response · Authors · 2022-11-14
> **Response to reviewer comments**
>
> Thank-you for the time and consideration put towards this review, we are pleased to hear that our motivation and methodology have been well conveyed. Regarding your other comments:
>
> W1: Future work. We agree that a model for method selection has additional value, however it is currently infeasible. Performance of each ANN method is dependent method hyperparameters as well as dataset properties. Training a model to optimise across these parameters is a long term goal for the field, but not currently feasible due to the intractable cost of sampling and evaluating points in the dataset property space. Current approaches for the relatively lower cost ANN on static datasets use the qualitative approach we follow in our paper.
>
> W2: Accepted. Thank-you for highlighting the missing citations, we will address this.
>
> W3: Accepted. Thank-you for highlighting the missing implementation details, we will clarify this in the manuscript. From the code documentation in the supplementary material - Online feature learning source \url{https://github.com/matsui528/deep1b_gt#bonus-deep1m} under MIT license and Online data collection source \url{http://corpus-texmex.irisa.fr/} under CC0 1.0 license.
>
> W4: Clarification. Our framework does include code to output these results. This includes the trade-off between each pair of measures and each individual measure over time, both of which can show comparisons between methods and between hyperparameter tunings of a single method. Currently they are not included in this paper due to the scale of the data and their sparse utility for a general audience. Are there specific use cases that you believe add value for the ICLR community?
>
> W5: Clarification. The methods evaluated in this paper do not update any internal parameters from query data or search performance. As such, each search is treated as an independent event and any mutual information will be ignored. This clarification will be added to the manuscript.

---

### Official Review · Reviewer_nVSk · 2022-10-26

**Confidence:** 5
**Correctness:** 4
**Technical Novelty And Significance:** 2
**Empirical Novelty And Significance:** 1
**Recommendation:** 3

**Clarity, Quality, Novelty And Reproducibility:**

The organization and clarity of the paper is excellent. The authors build their case carefully, showing why and how dynamic search cannot be accommodated by existing frameworks designed around static ANN search. However, the proposed framework can be regarded as a relatively straightforward extension of static benchmarking platforms. Although the paper does not give the full details of the implementation, the authors have stated their intention to make it freely accessible to practitioners.




**Strength And Weaknesses:**

Pros:

1) A thorough and compelling motivation is given for the work, in light of the severe computational and resource implications of dynamic search, particularly as regards the context of deep learning. The categorization provided for dynamic search is well-argued, and a good case is made for the need of a benchmarking framework for parameter tuning of ANN methods in dynamic settings.

2) The experimental evaluation does show that certain classes of ANN methods (in particular, graph-based methods) have greater prospects for handling high-dimensional and highly-dynamic data search problems, such as those that typically arrive in feature learning in ML/DL contexts.

3) The work seems ready for deployment in ML/DL development. The authors intend to make their framework openly accessible to practitioners wishing to automate the runtime automation of parameter-tuning for ANN search.

Cons:

1) Although the authors are careful to show a connection to ML/DL practice, the proposed framework is entirely generic as to the application - there are no unique issues that arise in learning contexts that require special treatment within the framework. As such, the topic is far from central to the interests of the ICLR community, and might be better suited to a different publication venue.

2) The authors do not demonstrate their framework on dynamic data sets of the same scale as is commonly encountered in learning.  In their experimentation, they consider only two datasets: one having 100,000 samples in 128 dimensions, with 100,000 added events, and the other having 5000 samples in 96 dimensions with 100,000 update events.

3) From the experiments, it is not clear whether `successful' parameter tuning is even capable of reducing the computational costs of ANN search methods enough to allow them to be used directly in deep learning. Practitioners and researchers often use fast heuristic methods such as minibatch sampling followed by brute-force search. Although the proposed framework can accommodate such heuristics, it is not clear that any of the more sophisticated ANN search methods would meet the time budgets available in practice, even when properly optimized.


**Summary Of The Paper:**

This paper is concerned with the problem of evaluating approximate nearest neighbor (ANN) search strategies for use in high-dimensional and highly-dynamic settings, such as in deep learning, where features must be learned from sample neighbors in embedding spaces.  The authors propose and discuss a characterization of dynamic search problems in terms of 12 measures such as search accuracy, runtime, initialization time, etc.  Based on their characterization, a framework is proposed for specifying, evaluating, and performance-tuning ANN methods on dynamic search problems.  They then demonstrate their framework by providing an experimental evaluation of 5 representative ANN methods on 2 dynamic search benchmark data sets (1 data collection benchmark, and 1 online feature learning benchmark).


**Summary Of The Review:**

Although the work targets a practical need in ML/DL, it is doubtful whether existing non-trivial ANN methods can meet practical time budgets even after parameter tuning. The work is not specific to learning, and many might regard this topic as only incidental to the interests of the ICLR community.

---

> ### Author Response · Authors · 2022-11-14
> **Response to reviewer comments**
>
> Thank-you for the time and consideration put towards this review, we are pleased to hear that our motivation, core results and code base successfully convey the purpose and function of this work. Regarding your other comments:
>
> Con 1: Clarification. Our online feature learning dataset provides a generalised but flexible model for ML/DL applications. It models a dataset that is converging through incremental updates. We use subsets of the 96 dimensional deep1B dataset (Efficient indexing of billion-scale datasets of deep descriptors, Babenko el al, CVPR, p2055-2063, 2016) with additional hyperparameters to simulate training epochs, batch size, convergence rate, a clustering coefficient and if groundtruth is measured after each batch or each epoch. This allows users to explore various ML/DL applications and can be further extended to suit more specific goals. This clarification will be added to the manuscript.
>
> Con 2: Clarification. We believe that 128 +/- 64 dimensions is typical for fully connected layers and auto encoder latent spaces. We also find that 100k events are sufficient to model dataset drift and to gauge the performance of ANN algorithms. Is there a specific application and magnitude that you believe adds additional value for the learning community?
>
> Con 3: Clarification. We simulate the effects of minibatching in our online feature learning dataset by computing groundtruths after each batch and each epoch. In practice, and in our evaluation, a recall-runtime tradeoff exists where minibatching is faster but unable to account for global changes in neighbourhood sets. The results presented in this paper compare against batch level groundtruths and plot the brute-force baseline accounting for speedups with various batch sizes of random samples. Smaller random batches are suitable for some applications that utilise local neighbourhood information, but are unable to utilise the quality of information provided by an accurate k-NN set. We demonstrate conditions where current ANN methods do improve upon bruteforce performance.

---

> > ### Comment · Reviewer_nVSk · 2022-12-07
> > **Response to the authors**
> >
> > I have read the authors' responses. While I appreciate their clarifications regarding the experimentation performed with batching (#3) and with regard to the sufficiency of the datasets for modeling dataset drift and gauging the performance of ANN (#2), they don't really address the point I had in mind. For (#3), scalability constraints can push implementers to consider minibatches of very small size, 100 to 1000, within which as few as 10 neighbors could be sought. For (#2), the experiments do not indicate that parameter tuning could reduce the costs of ANN by orders of magnitude, which is what would be needed to entice implementers away from low-cost heuristics within very small samples. While I believe that the authors' contributions have merit in situations where a full ANN search is to be computed across large datasets, I'm not convinced that it would have much benefit in DL.

---

### Decision · Program_Chairs · 2023-01-20

**Decision:**

Reject

**Justification For Why Not Higher Score:**

Limited relevance to ICLR and limited novel insights.

**Justification For Why Not Lower Score:**

Good quality work.

**Metareview: Summary, Strengths And Weaknesses:**

(a) Explains the issues involved in comparing algorithms for finding approximate nearest neighbors, and provides benchmarking code that measures numerous metrics.

(b) Thorough and sensible work.

(c) Only tangentially relevant to ICLR. Experiments are on small datasets not of clear practical relevance. Reviewers do not see new insights.

Comments by the AC: It would be good to include methods for finding exact nearest neighbors, since these can be sophisticated and effective. See for example "A new fast search algorithm for exact k-nearest neighbors based on optimal triangle-inequality-based check strategy" (2020) and "Fast nearest neighbor retrieval for bregman divergences" (2008). (Neither the AC nor any reviewer is an author of these papers.)

Although the datasets are dynamic, the experiments are static, i.e., offline from whatever the application would be. The authors would have more scope for novel insights if they provided an online tuning method. In other words, given one or more nearest neighbor algorithms, how  to optimize their performance adaptively as time goes by in the application? This adaptation could use bandit methods, Bayesian optimization, and/or more.

Every real-world applications is different and it is unclear which of several static benchmarks is most relevant. Therefore, and online tuning method would be more useful for applications.

**Summary Of Ac-Reviewer Meeting:**

No meeting.